# Soil Mite (Acari: Mesostigmata) Communities and Their Relationships with Some Environmental Variables in Experimental Grasslands from Bucegi Mountains in Romania

**DOI:** 10.3390/insects13030285

**Published:** 2022-03-14

**Authors:** Minodora Manu, Raluca Ioana Băncilă, Owen John Mountford, Teodor Maruşca, Vasile Adrian Blaj, Marilena Onete

**Affiliations:** 1Department of Ecology, Taxonomy and Nature Conservation, Institute of Biology Bucharest, Romanian Academy, Street Splaiul Independenţei, No. 296, 060031 Bucharest, Romania; 2Faculty of Natural Sciences, University Ovidius Constanţa, 900470 Constanţa, Romania; bancila_ralucaioana@yahoo.com; 3Department of Biospeleology and Soil Edaphobiology, “Emil Racoviţă” Institute of Speleology, Romanian Academy, 13 Septembrie Road, No. 13, 050711 Bucharest, Romania; 4Centre for Ecology and Hydrology, Maclean Building, Benson Lane, Crowmarsh Gifford, Wallingford, Oxfordshire OX10 8BB, UK; om@ceh.ac.uk; 5Grassland Research and Development Institute Braşov, Romania, Cucului Str., No. 5, 500128 Braşov, Romania; maruscat@yahoo.com (T.M.); blajadi@pajisti-grassland.ro (V.A.B.)

**Keywords:** abundance, dispersion, fertilizer, mite, richness

## Abstract

**Simple Summary:**

Grasslands are critical from ecological and pastoral points of view, being defined by valuable species of plants and animals. One of the most important biological components is soil fauna, as soil mites. The objective of the present study was to study the relation between fertilised experimental grasslands and soil mite (Acari: Mesostigmata) fauna. In this context, the aims of the research were to investigate the structural differences between mite communities and some key environmental variables from in five fertilised plots, the correlations of these variables with mites, and the dispersal rates of these invertebrate communities between grasslands. The number of species and their abundances recorded the highest values in the reseeded grassland and the lowest in the control plot. An indicator species analysis identified characteristic species for each experimental plot. The structural differences of the mite communities between plots were due to the significant influences of environmental variables. Between the experimental grasslands dominated by the accessory and accidental species, the dispersal rate of the mite communities was higher, in comparison were populations characterised by the constant species. Controlled and rationale use of agrochemicals (organic and chemic), influenced positive the numerical abundance and species richness of the soil mite communities, improving the soil environmental conditions for these invertebrates.

**Abstract:**

The main objective of the study was to analyse, for the first time in Romania, the relationships between five experimental grasslands and Mesostigmata fauna, considering: (1) the structural differences between mite communities; (2) the variations in some important abiotic factors (vegetation cover, soil temperature, soil moisture content, pH, soil resistance at penetration); and (3) the influence of these abiotic factors on the structures of Mesostigmata mite communities and the dispersal rates of these communities between the investigated plots. In total, 250 soil samples were analysed in July 2017, revealing the presence of 30 species, with 1163 individuals. Using the multivariate analysis, we observed that each experimental plot was defined by characteristic environmental conditions, i.e., vegetation cover, soil moisture content, and soil temperature differed significantly between the experimental grasslands. Each experimental plot was characterised by a specific indicator species and population parameters (numerical abundance and species richness). The effects of these soil variables were even demonstrated at species level: *Veigaia planicola*, *Geolaelaps nolli,* and *Gamasellodes insignis* were influenced by vegetation cover, *Lysigamasus conus* and *Dendrolaelaps*
*foveolatus* by soil temperature. The dispersal rates of mite communities from one plot to another were higher in the grasslands, where there were euconstant–constant species.

## 1. Introduction

Grasslands are some of the most important terrestrial biomes, with permanent grasslands representing 33% of the total utilised agricultural area in Europe and 18.90% in Romania [1,2]. Grasslands are among the most species-rich habitats in Europe [3,4,5,6,7,8]. The ecological goods and services provided by grasslands include forage and livestock, biodiversity, carbon storage, soil protection in ecosystems endangered by erosion and desertification, water management and purification, tourism, and recreation [9]. Mountain grasslands of Romania are critical from ecological and pastoral points of view. They are defined by valuable species of plants and animals that often have ecological plasticity [10,11,12,13,14,15]. These ecosystems are exposed to degradation processes through overgrazing, leading to changes in floristic composition, being dominated by *Nardus stricta*, a useless species for livestock, such as cows. This overgrazing took place after 1990 in the subalpine pastures of Romania [10,11] and was the reason, in 1995, for initiating experiments at the Mountain Grasslands Research Centre (MGRC) located in Blana Mountain–Bucegi Massif, Romania (belonging to the Grassland Research and Development Institute Braşov). These experiments tested the efficiency of various types of sward improvements applied to pastures degraded by *N. stricta,* as assessed by the evaluation of cow milk production. The types of sward improvements tested were fertilisation, calcium liming, paddocking, and reseeding. These improvement measures adjusted the soil chemical deficiencies, preserving grassland biodiversity and increasing their productivity [2,11]. According to Blaj et al. [16], the effects of these measures could be detected after 20 years of application if these areas are then included in a rational grazing system, from an agronomic perspective, with repeated applications of moderate rates of fertilisers.

Many researchers have shown that maintaining functionally diverse invertebrate communities in grasslands and agroecosystems may have positive effects on ecosystem functionality and in the provision of ecosystem services by increasing ecosystem coupling [17,18]. Among these invertebrate groups are soil mites (Acari: Mesostigmata). Most species are predators and about 11,000 species have been identified in the world [19]. In the soil, they participate indirectly in the decomposition process, feeding mainly on decomposer invertebrates with soft bodies (e.g., springtails, nematodes, larvae of insects, or oribatids) [20,21].

There are many studies throughout the world concerning the structures of mesostigmatid soil mite communities from various types of ecosystems in relation to different environmental variables [20,22,23,24]. These studies revealed that mite communities had spatial variabilities that change over time and that biotic and abiotic factors exert regulatory effects. Researchers highlighted that it is possible to predict the impact of habitat changes on specific soil organisms depending on their ecologies [20,24]. If we consider the research made on the Mesostigmata fauna in grassland ecosystems—these studies highlighted the influence of local and large-scale drivers on these invertebrate communities, but they were only conducted in a few countries in Europe, e.g., the U.K., Switzerland, Austria, Germany, Ireland, and Georgia [18,25,26,27,28,29,30,31,32,33], as well as elsewhere throughout the world, such as in New Zealand, Canada and Turkey [34,35,36,37,38,39]. Researchers studied the influences of grazing intensity, fertilisation, and different abiotic factors on the Mesostigmata communities. Other scientists researched the structural, seasonal, and vertical distributions of the mite populations from different types of grasslands [33,39,40]. In Romania, studies on mesostigmatid mites have focused on qualitative and quantitative data [41,42] or on the influences of heavy metal pollution [13,15] and other climatic factors on their communities [14]. None of these studies considered fertilised experimental grasslands and their relationships with mesostigmatid fauna. In this context, the aims of the present research were to investigate: (I) the structural differences between mite communities in five fertilised experimental plots; (II) the differences among certain key abiotic factors recorded in the five fertilised experimental plots; (III) the influences of these abiotic factors on the structures of mesostigmatid mite communities; and (IV) the dispersal rates of the mite communities between experimental plots.

## 2. Materials and Methods

### 2.1. Study Area

The research was carried out at the Mountain Grasslands Research Centre (MGRC) in the Bucegi Mountains, Romania, on gently sloping land situated at the base of Blana Peak (1875 m a.s.l.). The Bucegi Mountains are located on the east side of the Southern Carpathians, with an area of 300 km^2^. They have great structural and morphological complexities, with high peaks of altitudes between 1600 and 2500 m. The shape of the mountain range is that of an amphitheatre, with a wide opening on the south side (Figure 1).

The design at MGRC included a complex variety of measuring approaches within the experimental structure and was maintained during 1995–2016, with the following experimental plots:Control plot-grassland (CG) is a semi-natural pasture, unimproved, and grazed in the summer with cows. It is located at 45°21′23.7″ N; 25°27′38.9″ E; at 1758 m altitude, with an eastern exposure and a slope by 5°. The dominant plant species were *Agrostis capillaris* L., *Festuca ovina* L., *Pilosella aurantiaca* (L.) F.W. Schultz and Sch. Bip., *Ligusticum mutellina* (L.) Crantz, *Nardus stricta* L., *Potentilla aurea* L., *Polytrichum* sp., *Ranunculus acris* L., and *Trifolium repens* L. The control plot-grassland was chosen on the basis of having no inorganic or organic (cattle manure) fertilisers and no other human intervention (e.g., reseeding) (Figure 2a). Plot A is a semi-natural pasture, located at 45°21′24.3″ N; 25°27′39.5″ E; at 1758 m altitude, with an eastern exposure and a slope by 5°. The dominant plant species were *A. capillaris* L., *Deschampsia cespitosa* (L.) P. Beauv., *Festuca rubra* L., *Phleum alpinum* L., *Poa pratensis* L. and *Polygonum bistorta* L. It was fertilised with chemical fertilisers in three periods 2000–2002, 2010–2012, and 2014–2016, with an average application rate per year by 100 kg ha^−1^ N + 50 kg ha^−1^ P_2_O_5_ + 50 kg ha^−1^ K_2_O (Figure 2b).Plot B is a semi-natural pasture, located at 45°21′24.9″ N; 25°27′39.9″ E; at 1782 m altitude, with an eastern exposure, and no slope. The dominant plant species were *D. cespitosa* (L.) P. Beauv., *P. alpinum* L., *Phleum pratense* L., *R. acris* L., and *T. repens* L. It was chemically fertilised in 1996–1998 with an average application rate of 150 kg ha^−1^ N + 75 kg ha^−1^ P_2_O_5_ + 75 kg ha^−1^ K_2_O and organically fertilised in 2004, 2010, and 2016 by paddocking with dairy cows. Cattle manure was added as an input of organic matter (Figure 2c).Plot C is a semi-natural pasture, situated at 45°21′24.7″ N; 25°27′35.0″ E; at 1782 m altitude, with an eastern exposure and a slope by 5°. The dominant plant species were *Agrostis vinealis* Schreb., *L. mutellina* (L.) Crantz, *Poa annua* L., *P. pratensis* L., *P. bistorta* L., and *T. repens* L. It was limed in 1995, chemically fertilised similar to the B variant, and paddocked in 2003, 2009, and 2015. To correct the acidity of the soil to 2/3 Ah (hydrolytic acidity), a sterile lime powder (CaO) was used, in a dose of approximately 7.5 tons/ha, only once at the beginning of the experiments in the summer of 1995. In plot C, the powdered lime was spread on the surface of the land, without incorporation. Cattle manure was added as an input of organic matter (Figure 2d).Plot D is a pasture that was reseeded in the late summer of 1995, after a herbicide Roundup at 5 l ha^−1^, calcium liming (the CaO was incorporated in soil), being chemically fertilised, similar to the plot C variant, and paddocked in 2002, 2008, and 2014. It is situated at 45°21′23.9″ N; 25°27′34.5″ E; at 1784 m altitude, with an eastern exposure, and a slope by 5°. The sown mixture of grasses and perennial legumes comprised: *P. pratense* L. Favorit variety (40%), *F. pratensis* Huds. Transilvan variety (25%), *Lolium perenne* L. Marta variety (5%), *Trifolium hybridum* L.—local population from Braşov (15%), and *Lotus corniculatus* L. Livada variety (15%). Cattle manure was added as an input of organic matter (Figure 2e).

The dominant plant species were *Alchemilla vulgaris* L., *D. cespitosa* (L.) P. Beauv., *Holcus lanatus* L., *R. acris* L., *Taraxacum officinale* F.H. Wigg., and *T. repens* L. (Figure 1).

The plots were grazed for 85–90 days per year, by three Schwyz brown cows, fed exclusively with grass [16]. The type of soil in all experimental plots was podzol [43]. The area of a plot was 0.75 hectares. The shape of each investigated plot was a rectangle, with a length of 125 m and width of 60 m.

### 2.2. Mite Samples

Samples were collected one, in July 2017, in order to demonstrate the differences between structures of soil mite (Acari: Mesostigmata) communities from fertilised experimental grasslands. In total, 250 random soil samples were collected, with a MacFadyen soil core (diameter of 5 cm) at a 10 cm depth. In each plot, 50 soil samples were taken. The collected samples included the grass, moss and debris on the soil surface. The soil samples were collected randomly in order to catch the natural and heterogeneous distributions of the mite populations at the time of the sampling. The mites were extracted for 10–14 days using the Berlese–Tullgren method, using natural light and heat [44,45]. Due to the high number of samples collected from the field (250) and to the limited number of sample which could be sort one time (75 samples), some samples were kept in a refrigerator (at 4° C) until the following extraction (14 days). Taxonomic identification and counting were conducted using a Zeiss stereo-microscope and an Axio Scope A1 Zeiss microscope. Some of the mites were preserved in ethyl alcohol (75–90%) and others were dissected after clearing in lactic acid. Body parts were mounted in a polyvinyl alcohol–lactic acid mixture (PVA) medium [45].

The mites were identified to species level using the published identification keys [46,47,48,49,50,51,52,53,54,55]. The identified species are in the mite collection of the Institute of Biology Ecological Station at Posada.

### 2.3. Environmental Variables

At the same time as the collection of the mite samples, the following environmental variables were quantified: vegetation cover (VegCovr) (%), soil moisture content (H) (%), soil temperature (Ts) (°C), soil acidity (pH), and soil penetration resistance (RPs) (Map). The vegetation cover was determined using a pratological method, which took into consideration the percentage participation in the biomass of botanical components by economic groups (as grasses, legumes, mosses and lichens, wood species). It is one of the most recommended “fast” methods for determining grassland vegetation coverage [56]. The pH was measured with a C532 Jasco Consort pH-meter. Soil moisture content and temperature were measured with a digital thermohygrometer PCE-310. This digital thermohygrometer measures humidity and soil temperature using an optional external sensor. Resistance at penetrance was determined with a soil penetrometer, Step System GmbH, 41010. Soil Penetrometer (item 41010) allows you to measure and monitor the soil density at all stages of vegetation. In total, 50 soil samples per experimental plot (250 in total) were investigated for each environmental variable. The soil variables were measured at 10 cm depth.

### 2.4. Data Analysis

As the investigated plots were directly adjacent to each other (separated by wire fences), the dispersal rates of mite communities were considered. They were calculated using quantitative data (frequencies) according to studies developed by specialists [57,58] and applied on Mesostigmata mites [59]. Frequency was calculated as a ratio between the number of samples with an “a” species (p) and the total number of samples collected at the same time (*P*), as following: F = *p*/*P* *100 [60].

According to the frequency index (F), we classified the mites in the following groups: F5, euconstants (>50%), F4, constants (30.1–50%), F3, subconstants (15.1–30.0%), F2, accessory species (5.0–15.0%), and F1, accidentals (<5%).

The dispersal rate was measured using the following formula:

(DD4) = (2W/(A + B)) ((A − B)/(A + B − W)), where W represents the sum of the minimum frequency for the rarest species, and A and B are the sum of frequencies of all species in each studied ecosystems [59]. We considered the rarest species to be those classified as accessory (F2) and accident ones (F1).

The completeness of the mite species inventory was examined by visual inspection of Mao Tau species accumulation curves for each treatment and control, and of pooled data for all sites. We analysed three assemblage traits of the mite species: composition, abundance, and species richness.

Firstly, we investigated the species composition and its relationship with the environmental variables by applying constrained correspondence analyses (CCAs) to the mite species abundance matrix. A CCA can be used to model the multivariate response of a species assemblage to a matrix of environmental variables [58,61,62]. The species abundance matrix was ln (x + 1) transformed to maintain normal distribution and to avoid the “arch effect” in a CCA [63]. All numerical environmental variables were scaled to mean zero and unit variance. The CCA was performed using the vegan package [64]. To select the variables of the environmental matrix that best explained the species matrix (i.e., constraints), we performed a forward stepwise model using the “ordistep” function in the vegan package.

We calculated the variance inflation factors (VIFs) for each of the constraints and contrasts in the factor constrains from the environmental matrix using the function “vif.cca”. Values over 10 indicate redundant constraints. Using a stepwise model, all of the selected variables were independent from each other (VIF < 10). The permutation procedure (based on 9999 cycles) was used to test the significance of the CCA model of the explanatory variables in CCA and explain the variance by the CCA axes [64].

To examine mite community patterns in relation to treatment, we performed a non-metric multidimensional scaling (NMDS) ordination with a Bray–Curtis similarity index. To determine whether the treatments affect the mite communities, we used analysis of similarity (ANOSIM) with a Bray–Curtis similarity [65]. ANOSIM tests whether community differences among groups are significant by calculating a global R-statistic, which is an absolute measure of the distance between groups. An R-value close to 1 indicates strongly distinct assemblages, whereas an R-value approaching zero indicates that the assemblages are barely separable.

To identify the mites characteristic to the four treatments and control, we used the indicator value method [66]. The method assesses the specificity (uniqueness to a particular treatment) and fidelity (frequency of occurrence) of a species. A high indicator value (IndVal, expressed as a percentage) suggests that a species can be considered characteristic of a particular treatment. The IndVal for each species was calculated based on a species abundance matrix. To test for the significance of IndVal measures for each species, we used a random reallocation procedure of sites among the site groups [66]. Following this procedure, a species is considered characteristic to treatment type or control if the species IndVal is >25% and significant at *p* < 0.05 [66].

Abundance was expressed as the total number of individuals per site. General linear mixed models (GLMMs) were used to examine how mite abundance and species richness responded to treatment and environmental variables. In the models, we included treatment as a fixed factor, environmental variable as a covariate, and the site as a random factor. We assumed a Poisson error distribution and a log function. We designed a set of 15 candidate models. The first 11 candidate models included a single predictor. We also considered models that included two, three, and four predictors, with no collinearity problems. To detect collinearity between two or more predictor variables, we used VIF. A VIF > 3 was a signal of collinearity. We excluded highly collinear variables through “vifstep” in the “usdm” package. “vifstep” calculates VIF for all variables, excludes the one with the highest VIF, and repeats the procedure until there are no variables with a VIF greater than three remaining [67].

To assess the relative performance of these models, we performed model selection based on Akaike’s information criterion corrected for sample size [68,69], using the “aictab” function in R package AICcmodavg [70]. We ranked the models according to their AICc. The model with the lowest AICc was used as the reference for calculating the AICc difference (∆i). The relative evidence of each model was given by Akaike weights (wi). These were interpreted as the likelihood of a model given the data and the model. Models within two AICc units of the AICcmin were considered competitive and more plausible than others [68]. We used the most parsimonious model to obtain parameter estimates and predictions of abundance and species richness. The effects of other environmental covariates included in the best models were weak; thus, model averaging, which might be used to account for model selection uncertainty, would only shrink those effects [68]. Tukey tests were performed for multiple pairwise comparisons. All statistical procedures were implemented in R 4.0.2 [71].

## 3. Results

In over twenty years of the experiments, the presence of *Nardus stricta* in the vegetation layer decreased from a 60% cover at the beginning of experiments to 19% as a result of rational grazing with dairy cows (the plots were grazed for 85–90 days per year, by three Schwyz brown cows). In the limed and sown experimental plots, *N. stricta* has completely disappeared. Usage of chemical fertilisers stimulated the development of *Festuca nigrescens* and *A. capillaris*, while the liming stimulated *P. pratensis* and *T. repens*. 

None of the species’ accumulation curves (for each treatment) approached an asymptote (Figure 3). This indicated that species were correlated with the sampling effort and that more samples were required to detect all of the taxa theoretically expected per treatment.

The global CCA model, which selected VegCovr and Ts as meaningful explanatory variables, was significant (*p* = 0.004). The majority of this variation was explained by the first axis, only the first canonical axis being significant (first axis F = 2.747, *p* = 0.004; second axis F = 1.224, *p* = 0.228). The first axis accounted for 69.18% and the second for 30.82% of the total variation, respectively, and both explained 13.40% of the total inertia in the overall mite species data. We highlighted 29 species (r ≥ +/−0.5), which significantly correlated with one or both of the first two CCA axes. Amongst these, *Dendrolaelaps foveolatus* and *Lysigamasus conus* were strongly correlated with Ts, and *Veigaia planicola* and *Gamasellodes insignis* with VegCovr (Figure 4).

NMDS ordination of mite assemblages in relation to differences in treatments illustrates that there was a lot of overlap in mite assemblage compositions among the treatments (Figure 5). The ANOSIM tests indicated that the differences in mite communities among treatments were significant (R = 0.066, *p* = 0.003).

The indicator species analysis identified one characteristic species for each treatment A, B, A and C, C and D, and three species for treatment D (Table 1). The model selection indicated that for abundance and species richness, the best model supported by the mite data (ΔAICc < 2) (Table 2) included only treatment. The abundance (df = 4, chi square = 76. 661, *p* < 0.001) and the species richness (df = 4, chi square = 95. 325, *p* < 0.001) differed significantly among treatments.

The Tukey multiple comparisons test indicated that both abundance and species richness were significantly higher in all of the treatments compared to the control, except for treatment B (Table 3).

All of the environmental variables significantly differed among treatments, except the soil penetration resistance (Table 4).

The Tukey multiple comparisons of means showed significant differences between CG and A, B, and C treatments for vegetation cover, between CG and A, C, and D for soil moisture content, and between CG and D for soil temperature (Table 5).

In total, 30 species with 1163 individuals were identified in all investigated experimental plots. The highest values of species richness and numerical abundance were obtained in plot D (26 species, with 584 individuals) and the lowest in CG (5 species, with 50 individuals) and in B (7 species, with 52 individuals) (Figure 6a,b).

Taking account of the species compositions of the all investigated populations, we observed that the highest numerical density was obtained with *A**rctoseius cetratus, D. foveolatus, Geolaelaps praesternalis* and *L. conus* (Appendix A). Examining frequency, in all investigated areas, the euconstant species were *A. cetratus* and *Geolaelaps nolli*; the constant ones were *L. conus* and *Neopodocinum mrciacki*; subconstant ones were *Cheroseius bryophilus* and *Pergamasus laetus*; the remainder being accessory and accidental species (Appendix A). In plot CG, the dominant species was *A. cetratus*. No species was recorded as euconstant or constant, in the plot CG. This was demonstrated by the dispersal index, which showed traffic between B and C plots to the CG area (Appendix A). In plot A, the same situation was recorded: no euconstant or constant species, but the numerical dominants were *A. cetratus*, *G. nolli* and *Mixozercon sellnicki*. In plots B and C, the dominant species are *D. foveolatus* and *A. cetratus*. Only accessory and accidental mite species were identified in these grasslands. In plot D, the number of numerically dominant species was higher. Species such as *D. foveolatus*, *C. bryophyllus*, *G. nolli*, *L. conus* and *A. cetratus* were dominant. The *A. cetratus* was defined as euconstant and *L. conus*, constant (Appendix A).

With respect to the dispersal rates (which is based on the frequency classes), the highest values were obtained between the mite communities from the plots A–B (DD4 = 3.01), B–CG (DD4 = 2.41) and A–CG (DD4 = 1.03). The less mobile communities of mites were those grouped between plots from D–C (DD4 = 0.11), C–A (DD4 = 0.08) and C–B (DD4 = 0.93).

Five environmental variables were studied: vegetation cover, soil temperature, soil acidity, soil moisture content, and soil penetration resistance (Figure 6). The highest percentage cover of vegetation was recorded in plot A (75.96 ± 8.74 SD) and the lowest in plot C (57.28 ± 8.59 SD). For soil temperature, the highest value was obtained in plot C (17 °C ± 1.62 SD) and the lowest in plot D (14.7 °C ± 1.74 SD). The most acidic soil was found in plot B (4.55 ± 1.22 SD) and the lowest value of this variable was obtained in the soil of plot C (5.05 ± 0.42 SD). In plot A, the soil recorded the highest moisture content (69% ± 6.82 SD), with the lowest value from the soil from plot C (61.5% ± 5.9 SD). The highest value of soil resistance at penetration was obtained in plot B (1.39 MpA ± 0.14 SD) and the lowest in plot A (1.30 MpA ± 0.13 SD) (Figure 7).

## 4. Discussion

Comparison of the species richness (30 species) and numerical abundance (1163 individuals) in the five experimental plots produced similar results to those obtained in other geographical areas in Romania e.g., natural pastures on the Moldavian Plain (38 species), or polluted grasslands in Transylvania (9–28 species, with 28–267 individuals; 961 individuals in total) [13,14,41,42]. If we then compare our results with other natural pastures from Europe, species-richness and numerical abundance are at three levels: (a) similar results have been reported in Ireland and Norway (52–54 species); (b) higher species richness in Latvia (75 species); and (c) lower species-richness in studies from Ireland, Poland, Holland, Germany or Austria (20–50 species) [28,40,72,73,74,75,76]. The Romanian and European data are comparable, i.e. the soil samples were collected with a similar soil core (MacFadyen) and in the same type of grassland ecosystem.

Comparing soil mite communities within investigated grasslands and forest ecosystems from Bucegi Massif, the results revealed that species richness and numerical densities from the experimental plots were much lower than those obtained in *Abies alba* (80 species with 5369 individuals), *Fagus sylvatica* (73 species, with 6881 individuals) or *Picea abies* (68 species, with 11191 individuals) forests. The environmental conditions in the forest ecosystems (higher soil moisture contents, a lower soil temperature and a higher quantity of organic matter) and the lack of human interventions (all being natural, mature forests) are factors that contribute to these differences in structures of soil mite communities [77,78,79]. At the European level, and taking account of the same type of forest, the data obtained from fertilised grasslands are very different from those from deciduous forests (40 species of mites), or mixed forests (86 species with 814 individuals) and coniferous forests (45–84 species, with 660–1612 individuals) from Germany or Latvia (situated on northern part of our continent); but are similar to those obtained in pine forests (31 species with 461 individuals) or spruce forests (38 individuals with 650 species) from Poland (Central Europe) [21,72,75,80,81,82]. 

Comparison of species-richness and numerical abundance between the five experimental plots recorded the highest values in the reseeded grassland D and the lowest in the control CG. These differences were highlighted by the model selection. This variation in species richness and numerical abundance between experimental plots are due to the significant differences in environmental variables (according to Tukey multiple comparisons) between the grasslands with the exception of soil resistance at penetration. It is also possible that fertilisers influenced the structure of mite communities, since nitrogen input will produce a higher vegetation biomass (including dry biomass production), and at the same time modifies the microclimate of the grassland habitats, influencing water regime, microflora and soil structure [20,30,36,83]. In plot D, the dominant plant species (*F. rubra*, *H. lanatus*, *D. cespitosa*, *A. vulgaris*) are tall grassland species, which have a higher biomass than those from control plot CG (*P. aurea*, *R. acris*, *T. repens*). These tall plants from the reseeded plot are preferred by cows, have a high index of useful phytomass (6 from a maximum of 9) and hence organic matter input to the soil, after their ingestion by livestock, will be higher than in other experimental plots [10,11]. By applying fertilizers in the investigated experimental plots from Bucegi Mountains-Romania, the medium pastoral value increases by 29–41%, due to the effect of chemical fertilisation, 8–9% by paddocking and 13–14% by liming [16]. The pastoral value was measured using the Klapp-Elemberg percentage method [2,11]. Researchers from elsewhere in Europe highlighted that such tall-grass vegetation had a higher total springtail richness and mesostigmatid mite abundance than short-grass vegetation and also a different oribatid mite community composition. Although the abiotic and biotic variables differed between experimental plots, effects on soil mesostigmatid mites could be also explained by differences in litter quality [29,83]. 

We also observed that the dominant, as well as euconstant and constant species, in all the investigated plots were: *A. cetratus*, *D. foveolatus*, *G. praesternalis*, *G. nolli* and *L. conus*. *A. cetratus* has a great ability to colonise new environments (due to its phoretic abilities) and therefore, it is numerous in early stage succession. It is characterised by a high reproduction rate, short development time and tolerance to chemical contamination of soil [84]. *G. nolli* and *G. praesternalis* were species tolerating dry eco-systems, with soil poor in organic matter (e.g., polluted or natural grasslands, urban green areas, coastal habitats) [28,72,76,80,85,86]. The *D. foveolatus* occurs over a wide ecological range, being identified from industrial soil from Arctic areas as well as arable land and forest eco-systems in the temperate zone [13,14,15,81,87,88]. Focussing upon indicator species, we observed that *G. nolli* was characteristic of plot A, which was fertilised but at lower rates than other plots (except CG) and had the highest average values of vegetation cover, soil moisture content and the lowest average value of soil resistance at penetration. *P. norvegicus*, *C. bryophilus* were characteristic of plot D, where the herbicide Roundup was used, and where the soil temperature had the lowest recorded average value. *P. norvegicus* is a species characteristic of alpine areas, inhabiting cold habitats, and is resistant to the impact of agrochemicals [48,89]. *C. bryophilus* is found from the lowlands to the mountains, in litter, moss or even bark galleries [75]. The indicator species *M. sellnicki* was common within both plots A and C (i.e., fertilised grasslands) and showed wide ecological amplitude: from dry to wet soil and preferring mildly acidic soils (mean pH = 5.05). It has been described by other researcher as a characteristic species of dry meadows or high altitude grasslands in Slovakia, and in forest litter, soil and moss elsewhere in Europe [46,48,50,76,79]. *L. conus* was the common indicator species in plots C and D i.e., experimental grasslands were both CaO and organic fertilisers (cattle manure) were added. *L. conus* has been identified by other European acarologists in another type of experimental plot (TNT contaminated soils) or in reclaimed post-mining spoil-heaps and adjacent forest habitats [82,90].

In plots CG, A, B, C, no euconstant and constant species were recorded, but the highest rates of dispersal were observed. The NMDS ordination analysis revealed an overlap in mite assemblage composition among the plots. These values could indicate the mobility of the predatory mites (e.g., Mesostigmata), which are permanently looking for food, being in continuous interspecific competition. However, the mites are also looking for suitable environmental conditions (e.g., soil moisture content 60%, proper soil temperature between 21 °C and 24 °C, acid soil and high vegetation cover) [7,14,20,24,27,28,31]. In our study the environmental variables are not at all similar between plots. Significant differences in vegetation cover, soil moisture content and soil temperature were recorded between control plot grassland and the other plots.

Considering the effect of chemical treatments on the soil fauna of grassland eco-systems, some researchers have discovered that disturbance strongly reduced the abundance, diversity and species richness of oribatid mites (Oribatida) and springtails (Collembola), but had little effect on predatory mites (Mesostigmata) [26,37].

In the reseeded plot D, which differs principally from the other treatments through application of the glyphosate herbicide ‘Roundup’, one euconstant and one constant species were present (*A. cetratus* and *L. conus*). This plot had clearly lower values of dispersal rates in comparison with the other experimental plots. ‘Roundup’ is non-selective, killing almost all invertebrates above the ground, and has a relatively short environmental half-life (with an average by 30 days in temperate climate), owing to microbial degradation in the soil. Other studies have revealed that it persists in the soil of forest eco-systems as follows: 55 days in USA, 259–296 days in Finland, 335–336 days in Canada and 1–3 years in Sweden. In the soil of agroecosystems, it persists for about 249 days in Finland [91,92]. The fate of glyphosate depends on soil physicochemical properties (texture, organic matter content, pH), soil composition, its biological properties (microbial community), environmental conditions, the chemical properties of the herbicide, as well as the timing between precipitation and its application. Thus, in the present study, where the Roundup herbicide had been applied 20 years previously, it was clear that there was no residue of glyphosate in the soil. Other scientists studying soil fauna have observed that, following an initial decrease in numbers in response to herbicide application, the density of mites’ returns to the level present in untreated plots within several months [83,93]. However, application of good agricultural practices (cover crops; management of weeds, disease and pests; nutrient restoration; and a rational use of agrochemicals) in no-till fields (as the five investigated experimental plots from our study) increases the quality and quantity of litter, which constitutes suitable habitat for soil invertebrates, modifying faunal composition and abundance, by improving its resilience against insecticide application. The destruction of soil structure by soil tillage amplifies the effect of the herbicide on mites that are less affected in the no-tillage system despite application of the chemical. Other studies suggested that synergistic effects between soil physical or chemical disturbances may result in a higher impact upon soil fauna [94,95]. The plant residue cover/litter generated by the conventional agricultural practices contributed to increased numbers of litter invertebrates by generating an environment rich in food and with a suitable microclimatic. All vegetation biomass will be decomposed by the soil fauna (detritivore species such as oribatids, nematodes, springtails, etc.), which constitute favorable food for predatory mites such as mesostigmatids [19,21,45,94,96]. Plot D was the only experimental grassland where a mixture of grass and perennial legume seeds were sown, providing an additional input of nitrogen into the soil. According to McElroy [97], the transfer of nitrogen from legumes to grasses is an important process in low-input forage production systems, and may be improved by selecting compatible species and cultivars. From the ecological/pedological point of view, nitrogen is an important chemical element, with a key role in plant nutrition, but it is also a chemical element that is involved in organic matter decomposition, a process that is strongly correlated with certain soil fauna, including mites [19,45,94].

We observed that vegetation cover, soil moisture content and soil temperature differed significantly among the experimental grassland plots. Taking account of these differences between the investigated experimental plots, the five treatment plots could be characterised as follows: (1) plot A had the highest average values of vegetation cover, soil moisture content and the lowest average value of soil resistance at penetration; (2) plot B had the highest average value of soil resistance at penetration and the lowest average value of soil pH; (3) plot C had the highest average value of soil temperature and soil pH, the lowest average values of soil water content and vegetation coverage; (4) plot D had the lowest average value of soil temperature; and (5) plot CG had medium average values of all the investigated variables.

Using CCA multivariate analysis, we demonstrated that the influence of environmental variables is discernible even at the species level. The most significant abiotic factors were vegetation coverage and soil temperature. *V. planicola*, *G. nolli* and *G. insignis* were species influenced by the vegetation cover. According to an ecological study made on *Veigaia* species, *V. planicola* is an edaphic-detriticole species with a wide ecological distribution in Romania, from ultra-lowland up to the montane zone [85]. It is frequent and abundant in natural deciduous and coniferous forests and has been identified in small numbers in meadows and agroecosystems. It prefers dark soil with high humus content, but also occurs in mesobasic and mesotrophic soils [85]. This context helps explain the dependence of this species on the vegetation cover, since a higher percent cover of vegetation will offer this species a substrate richer in organic matter and a suitable microclimate. *G. nolli* is characteristic of wet meadows and is considered to be flood-resistant, but it is also found in newer fallow fields, being influenced by the habitat age and plant species [28,98,99]. In general, *G. insignis* prefers habitats rich in litter, humus and moss that are themselves correlated with the vegetation cover parameter [48,84].

*L. conus* and *D. foveolatus* were species influenced by the soil temperature. *D. foveolatus* occurs widely in diverse microhabitats, e.g., soil from agricultural and meadow ecosystems, compost, manure, forest litter and rotting bark, and is distributed especially in central Europe (Austria, Finland, Germany, Lithuania, Poland, Romania and Ukraine). In terms of soil temperature, other researchers have found it from industrially disturbed soils in the High Arctic, to the young forests after surface wildfire. Thus, its tolerance to the soil temperatures is quite high [15,48,86,100].

## 5. Conclusions

In this study, we analysed, for the first time in Romania, the structural differences between mite communities from five fertilised experimental grassland plots (including a control plot), examining the influences of some essential/basic abiotic factors (such as vegetation coverage, soil temperature, soil pH, soil moisture content, and soil resistance at penetration). In total, we analysed 250 soil samples, revealing the presence of 30 species of predatory mites (Mesostigmata), with 1163 individuals. Considering environmental variables, we demonstrated that vegetation cover, soil moisture content, and soil temperature differed significantly between the experimental grassland treatments. Each experimental plot was defined by characteristic environmental conditions, clearly showing distinguishable patterns.

Comparing the numerical abundance and the species richness between the five experimental plots, we observed that the highest values were obtained in grassland D and the lowest in CG. In the plots A, B, C the parameters had medium values. Each experimental plot was defined by specific indicator species. These species are characteristic of temperate grassland eco-systems, and some of them are resistant to the different chemical treatments of soil. The structural characteristics of mite communities, specific to each grassland, are due to the significant differences of environmental variables between plots, demonstrated through multivariate analysis. The effect of these variables was demonstrated even at species level: *V. planicola*, *G. nolli* and *G. insignis* were influenced by vegetation cover while *L. conus* and *D. foveolatus* are affected by soil temperature. From the results of the present study, we showed that the controlled and informed use of agrochemicals (after a period of six year from the last chemical treatment) improves the soil environmental conditions for mites. 

Analysing the dispersal rates of mite communities between the plots (based on species frequency), we demonstrated that, between the experimental grasslands dominated by the accessory and accidental species, the mobilities of these invertebrate communities were higher (between A–B–CG) in comparison with those from plots D and C, where these populations were more stable.

These results highlight that the proposed four objectives of the present study were accomplished. On the other hand, we realise that the study should continue, with more detailed investigations into the influence of other soil chemical variables on mite communities; this task will be considered as a future objective. This study demonstrates the importance of understanding the ecology of mite communities as an important tool in environmental assessment.

## Figures and Tables

**Figure 1 insects-13-00285-f001:**
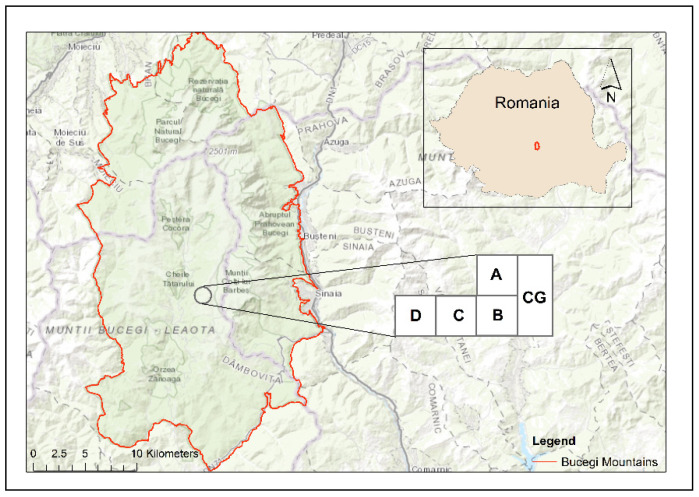
Geographical description of the investigated experimental plots from grassland ecosystems, Bucegi Mountains, Romania, 2017. Map created using ArcGIS software by Esri. ArcGIS and ArcMap are the intellectual property of Esri and are used herein under license. Version number: 10.4.0554. Copyright Esri. All rights reserved. For more information about Esri software, please visit www.esri.com (accessed on 27 October 2021). Base-map service layer credits: Esri, HERE, DeLorme, Intermap, increment P Corp., GEBCO, USGS, FAO, NPS, NRCAN, GeoBase, IGN, Kadaster NL, Ordnance Survey, Esri Japan, METI, Esri China, swisstopo, MapmyIndia, OpenStreetMap contributors, and the GIS.

**Figure 2 insects-13-00285-f002:**
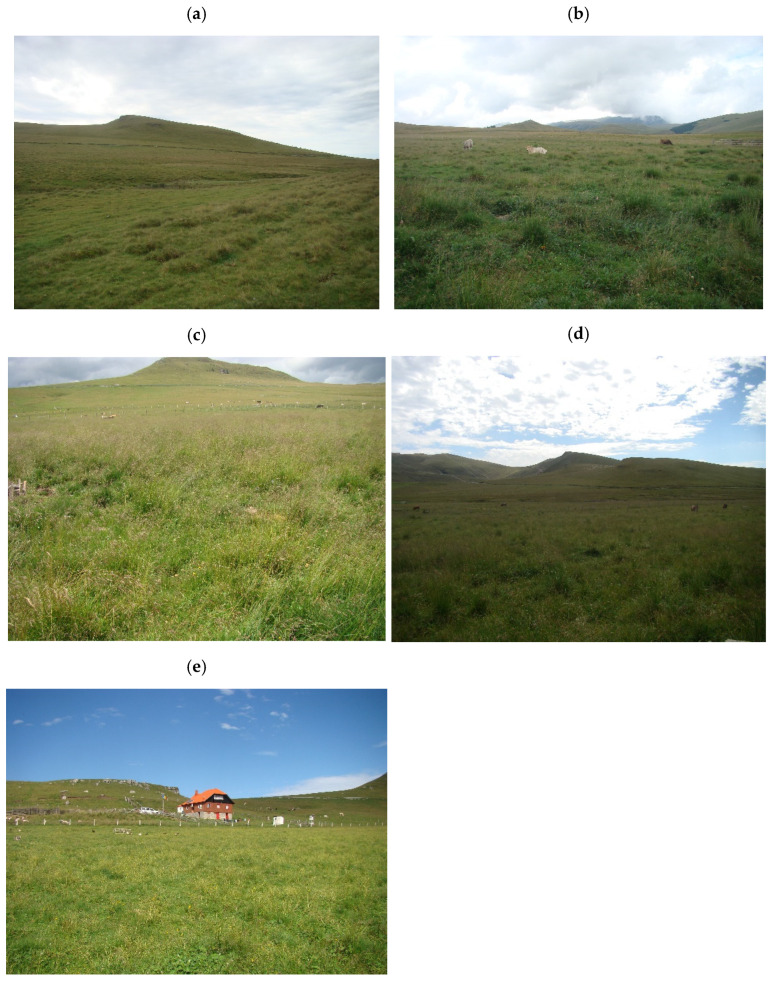
The general aspects of the investigated experimental plots from grassland ecosystems, Bucegi Mountains, Romania, 2017 ((**a**) control plot-grassland CG; (**b**) plot A; (**c**) plot B; (**d**) plot C; (**e**) plot D).

**Figure 3 insects-13-00285-f003:**
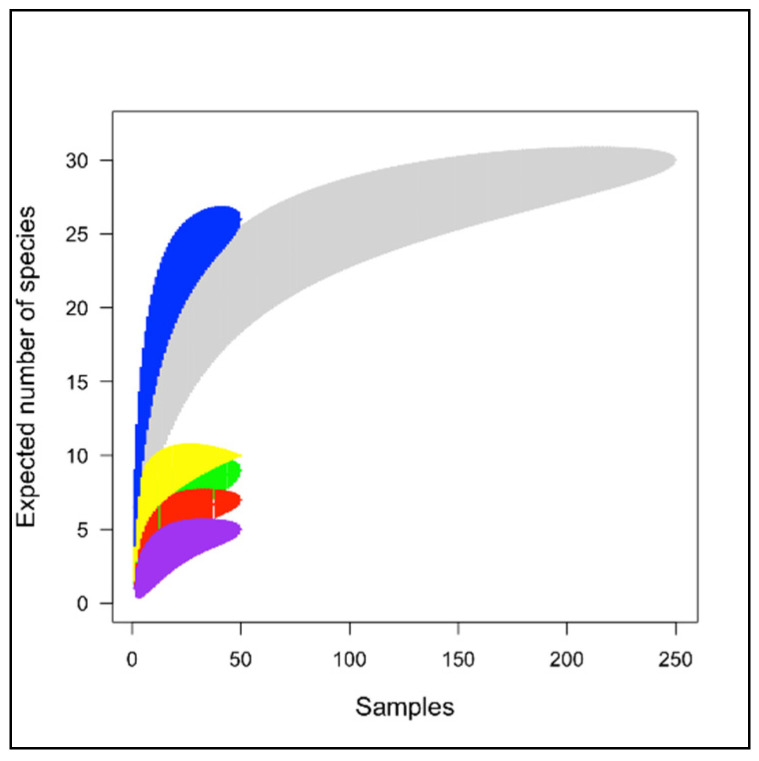
Mao Tau species accumulation curve per treatment: A (green), B (red), C (yellow), D (blue), CG (purple), and for all sites (grey).

**Figure 4 insects-13-00285-f004:**
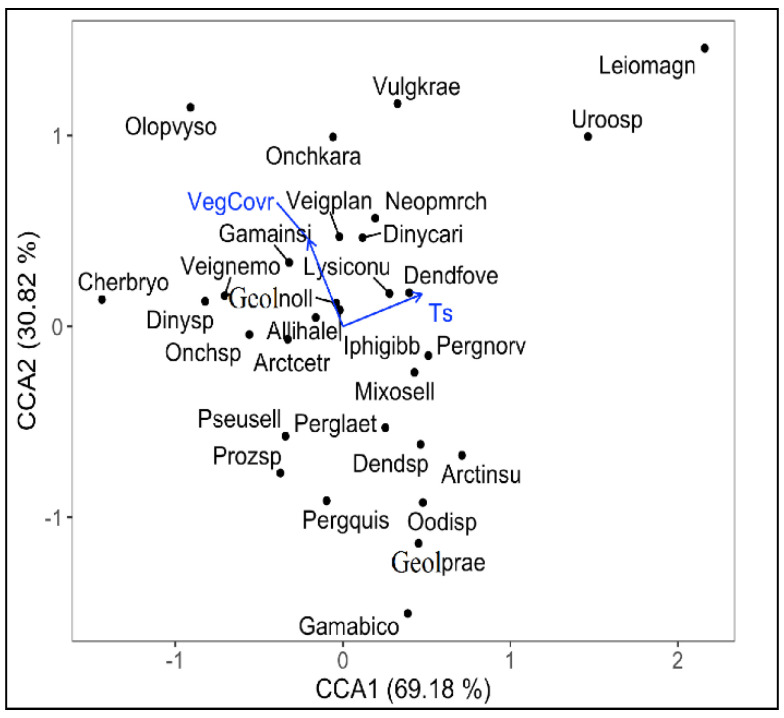
Biplots of the CCA model of the mite species abundance matrix in relation to environmental variables; soil moisture content—H (%); soil temperature—T soil (°C); soil pH; resistance of soil at penetrance—Penetr (Mpa); vegetation coverage (%). Species names were abbreviated using the first four letters of the genus and four letters of species name, respectively. Abbreviations for the species are shown in Appendix A.

**Figure 5 insects-13-00285-f005:**
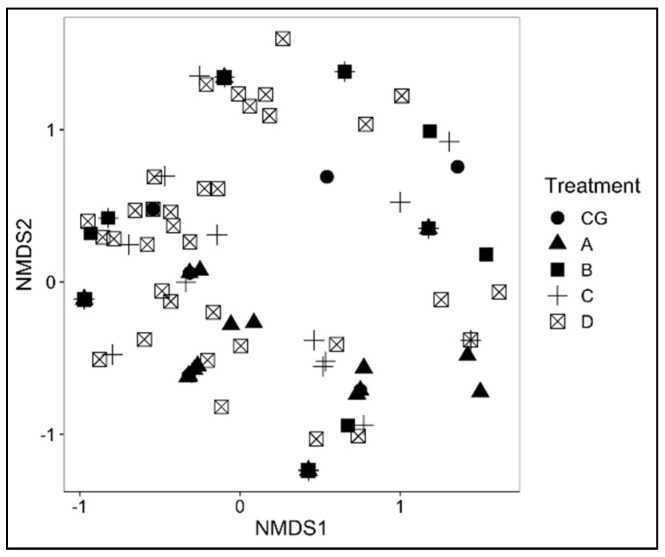
Non-metric multidimensional scaling (NMSD) ordination plots of mite species (stress = 0.273) assemblage-based treatments.

**Figure 6 insects-13-00285-f006:**
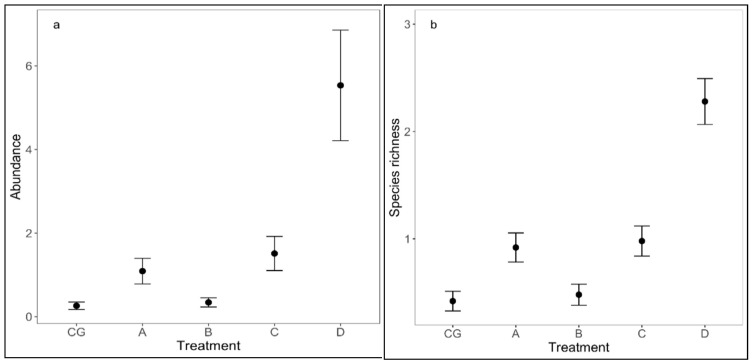
The effect of treatment on abundance (**a**) and species richness (**b**) of mites. The lines represent the standard errors.

**Figure 7 insects-13-00285-f007:**
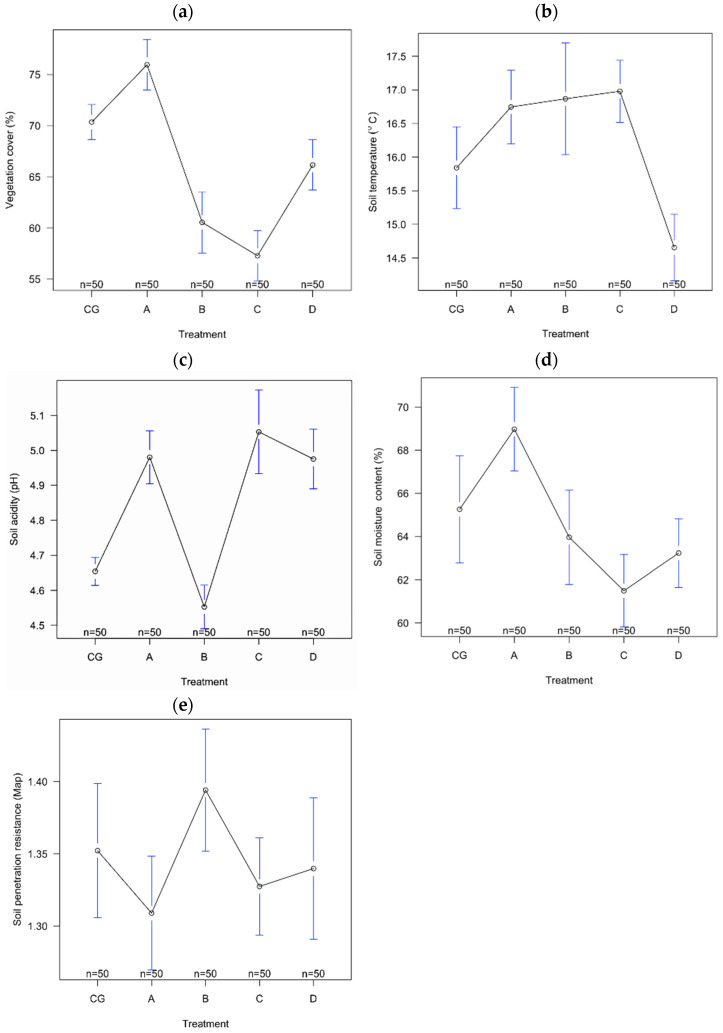
The variation of investigated environmental variables ((**a**) vegetation cover, (**b**) soil temperature, (**c**) soil acidity, (**d**) soil moisture content, (**e**) soil penetration resistance) among treatments of experimental grasslands from Bucegi Mountains, Romania. The open circles represent treatment means and the blue lines represent the confidence intervals.

**Table 1 insects-13-00285-t001:** Species identified as characteristics of treatments. IndVal = Indicator Value (%).

Treatment	Species	IndVal	*p*
A	*Geolaelaps nolli*	26.8	<0.05
B	*Dendrolaelaps* sp.	24.5	<0.05
D	*Dinychus* sp.	31.7	<0.01
	*Pergamasus norvegicus*	29.0	<0.05
	*Cheroseius bryophilus*	25.8	<0.05
A + C	*Mixozercon sellnicki*	28.1	<0.05
C + D	*Lysigamasus conus*	27.6	<0.05

**Table 2 insects-13-00285-t002:** Model selection results. Models are ranked in a decreasing order of the Akaike weights (wi). For clarity, models with wi < 0.02 are not shown. Statistics include LL—log likelihood; K—number of parameters; the second-order Akaike information criterion corrected for small sample sizes AICc; ∆i-AICc differences; wi-Akaike weights.

Model Structure	LL	K	AICc	∆_i_	w_i_
Abundance					
Treatment	−545.63	6	1103.26	0.00	0.27
Treatment + RPs	−545.47	7	1104.93	1.67	0.12
Treatment + VegCovr	−545.48	7	1104.96	1.70	0.12
Treatment + Rhs	−545.52	7	1105.04	1.78	0.11
Treatment + pH	−545.54	7	1105.09	1.83	0.11
Treatment + Ts	−545.60	7	1105.21	1.95	0.10
Treatment + VegCovr + pH	−545.40	8	1106.79	3.53	0.05
Treatment + VegCovr + Ts	−545.47	8	1106.94	3.68	0.04
Species richness					
Treatment	−305.74	6	623.47	0.00	0.26
Treatment + RPs	−305.36	7	624.71	1.24	0.14
Treatment + Ts	−305.48	7	624.97	1.50	0.12
Treatment + pH	−305.61	7	625.21	1.74	0.10
Treatment + Rhs	−305.71	7	625.43	1.95	0.10
Treatment + VegCovr	−305.74	7	625.47	2.00	0.10
Treatment + VegCovr + Ts	−305.48	8	626.97	3.50	0.05
Treatment + VegCovr + pH	−305.61	8	627.21	3.74	0.04

**Table 3 insects-13-00285-t003:** Results of multiple comparisons of abundance and species richness among treatments.

	Estimate	*SE*	z Value	*p*
Abundance				
CG–A	−1.432	0.421	−3.399	0.0027
CG–B	−0.272	0.446	−0.610	0.767
CG–C	−1.758	0.417	−4.216	<0.001
CG–D	−3.056	0.410	−7.451	<0.001
Species richness				
CG–A	−0.784	0.263	−2.977	0.015
CG–B	−0.134	0.299	−0.447	1.000
CG–C	−0.847	0.261	−3.249	0.007
CG–D	−1.692	0.237	−7.124	<0.001

**Table 4 insects-13-00285-t004:** One-way ANOVA results of environmental variables. df = degrees of freedom; MS = mean squared; F = F statistic; *p* = level of significance.

Environmental Variables	df	MS	F	*p*
Vegetation cover (%)	4	2800.9	37.66	<0.001
Soil moisture content (%)	4	2.504	30.91	<0.001
Soil temperature (°C)	4	48.29	10.79	<0.001
Soil acidity (pH)	4	394.2	7.951	<0.001
Soil penetration resistance (Map)	4	0.05135	2.29	>0.05

**Table 5 insects-13-00285-t005:** Results of Tukey multiple comparisons of the means of the environmental variables among treatments. diff = difference between means of the two treatments; LCL = lower confidence interval; ULC = upper confidence interval; *p* adj = *p*-value after adjustment for the multiple comparisons.

	Diff	95% LCL	95% UCL	*p* adj
Vegetation cover (%)				
CG–A	5.60	−0.859	10.340	0.011
CG–B	−9.82	−14.560	−5.079	<0.001
CG–C	−13.08	−17.820	−8.339	<0.001
CG–D	−4.20	−8.940	0.540	0.109
Soil moisture content (%)				
CG–A	0.326	0.169	0.482	0.011
CG–B	−0.101	−0.257	0.055	0.388
CG–C	0.399	0.242	0.555	<0.001
CG–D	0.321	0.164	0.477	<0.001
Soil temperature (°C)				
CG–A	0.904	−0.258	2.066	0.208
CG–B	1.026	−0.136	2.188	0.112
CG–C	1.138	−0.024	2.300	0.058
CG–D	−1.186	−2.348	−0.023	0.043
Soil acidity (pH)				
CG–A	3.716	−0.154	7.586	0.066
CG–B	−1.296	−5.166	2.574	0.888
CG–C	−3.770	−7.640	0.100	0.060
CG–D	−2.030	−5.900	1.840	0.601

## Data Availability

Data are contained within the article or Appendix A.

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
