# Peer review of "Soil Mite (Acari: Mesostigmata) Communities and Their Relationships with Some Environmental Variables in Experimental Grasslands from Bucegi Mountains in Romania"

_insects, 2022, doi:10.3390/insects13030285_

Round 1
Reviewer 1 Report
This is important work from a scientific point of view and necessary from a practical point of view. Mites are good bioindicators of environmental changes, hence they can be used in effective nature protection.
I accept the author's point of view on the analysis of the collected material and therefore I do not see the need to suggest other statistical analyzes.
Below are some suggestions that I think are worth taking into account.
- Can be consider posting photos of the sample collection sites. This will allow the reader to imagine the environmental and landscape conditions.
- Research on Mesostigmata mites in the Bucegi Mountains in Romania has already been carried out.
Stanescu M., Gwiazdowicz D.J. 2004. Preliminary research on Mesostigmata mites (Acari) from a spruce forest in the Bucegi Massif in Romania. Acta Scientiarum Polonorum Silvarum Colendarum Ratio et Industria Lignaria, 3 (2): 79-84.
It is worth referring to it in the discussion, comparing the results and the structure of mite communities (forest- grassland).
- The names of the mite species must be verified to use those currently in force eg. Geolaelaps nolli (Karg, 1962), Gaeolaelaps praesternalis (Willmann, 1949), no Hypoaspis. By the way line: 422 is preasternalis, shoud be praesternalis.
Author Response
REVIEWER 1
- Can be considering posting photos of the sample collection sites. This will allow the reader to imagine the environmental and landscape conditions.
Author answer: Photos of the sampling collection sites were added. Please, see the figure 2a, b, c, d, e from the manuscript!
- Research on Mesostigmata mites in the Bucegi Mountains in Romania has already been carried out.
Stanescu M., Gwiazdowicz D.J. 2004. Preliminary research on Mesostigmata mites (Acari) from a spruce forest in the Bucegi Massif in Romania. Acta Scientiarum Polonorum Silvarum Colendarum Ratio et Industria Lignaria, 3 (2): 79-84.
Author answer: I am agree with the reviewer, but in the present article the investigated ecosystems are different in comparison with previous article (there are grasslands and not spruce forest), even if the geographical location is the same. The Bucegi Mountains are spread on a huge area of 300 km2. On this area there are a lot of unstudied ecosystems from acarological point of view!
- It is worth referring to it in the discussion, comparing the results and the structure of mite communities (forest- grassland).
Author answer: At Chapter 4. Discussions the second paragraph was inserted: “Making a comparison between soil mite communities from investigated grasslands and some forest ecosystems from Bucegi Massif, the results revealing that the species richness and numerical densities from the experimental plots is much lower than those obtained in Abies alba (80 species with 5369 individuals), Fagus sylvatica (73 species, with 6881 individuals) or Picea abies (68 species, with 11191 individuals) forests. The specifically environmental conditions from the forest ecosystems (higher soil moisture contents, a lower soil temperature and a higher quantity of organic matter) and the lack of human interventions (all being natural, mature forests) are factors that contribute to this difference on structures of soil mite communities [72, 73, 74]. On European level, and taking account of the same type of forest, the obtained data from fertilized grasslands are very different from those from deciduous forests (40 species of mites), or mixed forests (86 species with 814 individuals) and coniferous forests (45-84 species, with 660-1612 individuals) from Germany or Latvia (situated on northern part of our continent); but are similar with those obtained in the pine forests (31 species with 461 individuals) or spruce forests (38 individuals with 650 species) from Poland (on Central Europe) [21, 76, 79, 83, 85, 90].”
- The names of the mite species must be verified to use those currently in force eg. Geolaelaps nolli (Karg, 1962), Gaeolaelaps praesternalis (Willmann, 1949), no By the way line: 422 is preasternalis, shoud be praesternalis.
Author answer: The scientific names of the species were changed in the manuscript, in the figure 3 and in supplementary material. Thank you very much for corrections!
Thank you very much for your precious advices and for your effort!

Reviewer 2 Report
Dear authors,
The manuscript is worth publishing in the journal; however, the corrections and improvements to be made according to my comments will make stronger it.
Regards,
---
My queries-suggestions
- In the author list, I could not see an expert acarologist at this group. To what degree familiar are they with the mesostigmat mite species?
- The selection criteria of the control station in the text should be explained.
- Sampling has been done only in July. Why? Seasonal changes of the mite communities could be determined by taking samples at different periods. Please explain.
- Why were only soil samples taken? These mites live in many habitats such as in moss and grass litter.
- When sampling, were samples of grass and debris on the soil surface taken or removed?
- Species identified as characteristics: The abbreviation ''spp.'' used as plural shows possibility of mixture of distinct unspecified species of the genus. It is not clear which one is characteristic in this case. If they were shown and analysed as "sp.1", "sp.2" etc. instead of "spp.", it could be understood which one is characteristic.

Author Response
REVIEWER 2
My queries-suggestions
- In the author list, I could not see an expert acarologist at this group. To what degree familiar are they with the mesostigmata mite species?
Author answer: The present study was based on team work, as we declared in the authors contributions. The main expert of Mesostigmata mites is Manu Minodora, which is expert for more than 20 years in this invertebrate group, being able to determine till species level.
- The selection criteria of the control station in the text should be explained.
Author answer: In the manuscript the following text was added:” The control plot-grassland was choose, taking into consideration that no chemical or organical (cattle manure) fertilizers were used and no other human intervention was recorded (as reseed).”
- Sampling has been done only in July. Why? Seasonal changes of the mite communities could be determined by taking samples at different periods. Please explain.
Author answer: In general, I am agree with the reviewer, but the present study wants to demonstrate the differences between mite communities structure from five experimental grasslands, on one time period, based on a high number of samples. We could compare this study with radiography at a certain moment of time (in our case July, 2017).
In the manuscript at 2.2 Mite samples- subchapter the following text was added: “In order to demonstrate the differences between structures of soil mite (Acari: Mesositgmata) communities from fertilized experimental grasslands, in the vegetation period, the samples were collected only one time.”
- Why were only soil samples taken? These mites live in many habitats such as in moss and grass litter.
Author answer: The samples were collected including the grass, moss and debris on the soil surface. This phrase was added in the manuscript at 2.2 Mite samples-subchapters.
- When sampling, were samples of grass and debris on the soil surface taken or removed?
Author answer: No, the soil surface wasn’t taken or removed. The samples were collected including the grass, moss and debris on the soil surface. This phrase was added in the manuscript at 2.2 Mite samples-subchapters.
- Species identified as characteristics: The abbreviation ''spp.'' used as plural shows possibility of mixture of distinct unspecified species of the genus. It is not clear which one is characteristic in this case. If they were shown and analysed as "sp.1", "sp.2" etc. instead of "spp.", it could be understood which one is characteristic.
Author answer: I am agreeing with the reviewer and thank you for your advice. In the manuscript, as well as in the supplementary material the spp. was changes with sp. (Dendrolaelaps sp. and Dinychus sp.).
Thank you very much for your precious advices and for your effort!
